# Higher-Level Steatosis Is Associated with a Greater Decrease in Metabolic Dysfunction-Associated Steatoic Liver Disease after Eight Weeks of a Very Low-Calorie Ketogenic Diet (VLCKD) in Subjects Affected by Overweight and Obesity

**DOI:** 10.3390/nu16060874

**Published:** 2024-03-18

**Authors:** Annamaria Sila, Sara De Nucci, Caterina Bonfiglio, Vincenza Di Stasi, Nicole Cerabino, Martina Di Chito, Roberta Rinaldi, Paola Cantalice, Endrit Shahini, Vito Giannuzzi, Pasqua Letizia Pesole, Sergio Coletta, Nicoletta Maria Tutino, Giuseppina Piazzolla, Raffaele Cozzolongo, Gianluigi Giannelli, Giovanni De Pergola

**Affiliations:** 1Center of Nutrition for the Research and the Care of Obesity and Metabolic Diseases, National Institute of Gastroenterology IRCCS “Saverio de Bellis”, Castellana Grotte, 70013 Bari, Italy; annamaria.sila@irccsdebellis.it (A.S.); sara.denucci@irccsdebellis.it (S.D.N.); vincenza.distasi@irccsdebellis.it (V.D.S.); roberta.rinaldi@irccsdebellis.it (R.R.);; 2Laboratory of Epidemiology and Statistics, National Institute of Gastroenterology “Saverio de Bellis”, IRCCS Hospital, Castellana Grotte, 70013 Bari, Italy; catia.bonfiglio@irccsdebellis.it; 3Department of Gastroenterology, National Institute of Gastroenterology “Saverio de Bellis”, IRCCS Hospital, Castellana Grotte, 70013 Bari, Italy; endrit.shahini@irccsdebellis.it (E.S.);; 4Core Facility Biobank, National Institute of Gastroenterology “Saverio de Bellis”, IRCCS Hospital, Castellana Grotte, 70013 Bari, Italysergio.coletta@irccsdebellis.it (S.C.); 5Laboratory of Clinical Pathology, National Institute of Gastroenterology “Saverio de Bellis”, IRCCS Hospital, Castellana Grotte, 70013 Bari, Italy; nicoletta.tutino@irccsdebellis.it; 6Interdisciplinary Department of Medicine, School of Medicine, University of Bari “Aldo Moro”, Piazza Giulio Cesare 11, 70124 Bari, Italy; giuseppina.piazzolla@uniba.it; 7Scientific Direction, National Institute of Gastroenterology “Saverio de Bellis”, IRCCS Hospital, Castellana Grotte, 70013 Bari, Italy; gianluigi.giannelli@irccsdebellis.it

**Keywords:** metabolic dysfunction-associated steatotic liver disease (MASLD), very low-calorie ketogenic diet (VLCKD), obesity, transient elastography (FibroScan)

## Abstract

The most common form of chronic liver disease, recently defined as MASLD, is strongly linked to obesity and metabolic syndrome. Lifestyle changes are part of MASLD prevention. The very low-calorie ketogenic diet (VLCKD) is a useful option for treating MASLD and reducing liver steatosis in patients with obesity. We assessed whether a greater degree of steatosis could have a positive or negative impact on how well 8 weeks of using the VLCKD improve steatosis and fibrosis in a patient population of overweight and obese individuals. Anthropometric parameters, along with changes in hormone and metabolic biomarkers, were also assessed both before and after the dietary change. The study population included 111 overweight (14.41%) or obese subjects (85.59%) aged between 18 and 64 years; the 75 women and 36 men involved were not taking any medicine. In both the raw (0.37 95% CI 0.21; 0.52) and the multivariate models (*model a*: 0.439 95% CI 0.26; 0.62; *model b*: 0.437 95% CI 0.25; 0.63), there was a positive and statistically significant correlation between the CAP delta value and the CAP before using the VLCKD. Additionally, the liver stiffness delta was found to be positively and statistically significantly correlated with liver stiffness before the use of the VLCKD in both models: the multivariate model (*model a*: 0.560 95% CI 0.40; 0.71; *model b*: 0.498 95% CI 0.34; 0.65) and the raw model (0.52 95% CI 0.39; 0.65). Systolic and diastolic blood pressure, insulin resistance (measured by HOMA-IR), insulin, HbA1c, fasting blood glucose, total cholesterol, LDL cholesterol, HDL cholesterol and triglycerides, BMI, waist circumference, and fat mass, were all decreased (*p* < 0.001) following the use of the VLCKD. However, following the use of the VLCKD, there was an increase in vitamin D levels. (*p* < 0.001). We found that using the VLCKD for 8 weeks has a greater effect on improving steatosis and fibrosis in subjects who initially have more severe forms of these conditions.

## 1. Background

Fatty liver disease is currently the main cause of liver disease, possibly because of the metabolic syndrome’s rising incidence. Metabolic dysfunction-associated steatotic liver disease (MASLD) currently affects over 25% of adults in Europe and 30% of people worldwide [1], making it the most prevalent type of chronic liver disease in developed nations [2]. MASLD occurs more often, reaching a prevalence of more than 70%, among subjects with metabolic diseases such as obesity, type 2 diabetes, and metabolic syndrome [3,4]. In addition to genetic factors and exposure to environmental risk factors, the main causes of metabolic dysfunction-associated steatotic liver disease (MASLD) are passive lifestyle habits, a hypercaloric diet, and insulin resistance (IR) [5,6]. Abdominal obesity, type 2 diabetes mellitus, low high-density lipoproteins (HDL), and hypertriglyceridemia and hyperuricemia are the main risks linked to MASLD [7,8,9]. In particular, a recent cross-sectional study has shown that obesity-induced hepatic steatosis is partly mediated by visceral fat accumulation [10]. When hepatic fat deposition is 5% or higher and is unrelated to viral infections, excessive alcohol consumption, or drug use, it is commonly referred to as hepatic steatosis [11,12]. NASH is a potential progression of nonalcoholic fatty liver disease (NAFLD), where fat accumulation is linked to hepatocyte inflammation with or without fibrosis [13]. Nonalcoholic steatohepatitis (NASH) is the second biggest cause of liver transplantations in the United States [14]. In addition to developing diabetes, cardiovascular disease, and kidney disease, patients with fatty livers are at risk of developing cirrhosis and liver cancer [15], all of which have a major negative impact on one’s quality of life. The terms NAFLD and NASH have been eliminated by the American Association for the Study of Liver Diseases (AASLD) due to their stigmatizing nature and potential for confounding [16]. This term was replaced with metabolic dysfunction-associated steatotic liver disease (MASLD), which refers to the presence of at least one of the five cardiometabolic risk factors defining the presence of metabolic syndrome. As a result, the term steatotic liver disease (SLD) was chosen because it encompasses various aetiologies of steatosis [16]. Nevertheless, there are insufficient data to determine whether MASLD is useful in detecting metabolic abnormalities and advanced fibrosis, despite some studies’ suggestions to that effect [17].

Although liver biopsy is the gold standard for MASLD diagnosis, staging, and patient management, it is too invasive to be used frequently in clinical practice. Proton magnetic resonance spectroscopy (1 H-MRS) is considered as the most accurate measure of hepatic triglyceride content (HTGC) and is accordant with liver biopsy in assessing and categorizing MASLD [18]. Although repeated exams can be carried out without concerns regarding safety, it is a costly procedure and time-consuming, and it is available only in a few specialized centers worldwide. 

One well-known ultrasound-based technique for determining the degree of hepatic steatosis is transient elastography, or FibroScan [19,20]. MASLD prevention includes lifestyle changes such as a healthy diet and increased physical exercise. These modifications improve IR, lower systemic inflammation, encourage weight loss and the reduction of body fat, and increase skeletal muscle mass [21]. 

“Low-grade” inflammation is the term used to describe the inflammatory state linked to obesity; it results from the down-regulation of anti-inflammatory cytokines and increased synthesis of pro-inflammatory molecules [22]. Low-grade chronic inflammation is one of the risk factors for MASLD that may be involved in the development of liver damage to cirrhosis [23].

MASLD is reversible through the correction of causative factors and implementation of lifestyle modifications when this hepatic disease is diagnosed at an early stage [24].

According to current recommendations, losing weight is the most effective way to treat inflammation. However, it is also important to keep in mind that high-quality diets, like the DASH diet and the Mediterranean diet, have been shown to effectively reduce liver steatosis and general inflammation [25,26]. 

Internationally recognized as a successful weight loss intervention, the ketogenic diet (KD) is typified by a sharp decrease in carbohydrate intake. Specifically, a very low-calorie ketogenic diet (VLCKD) is frequently regarded as a safe and effective therapeutic intervention for individuals with obesity [27,28,29,30,31]. The rapid mobilization of fat from adipose tissue and the weight loss resulting from a VLCKD offers a very useful option for decreasing liver steatosis and treating MASLD [31,32,33]. Remarkably, our earlier research employing VLCKD for eight weeks revealed a drop in circulating TNF-α levels and an increase in IL-10 concentrations, indicating the inhibitory effect of the VLCKD on systemic inflammation, which could potentially be linked to MASLD [34]. According to Watanabe et al., individuals with insulin resistance benefit more from a VLCKD than those with less IR in terms of improving MASLD, indicating that patients with IR should receive strong recommendations for this dietary intervention [35]. However, these authors did not evaluate whether the grade of basal liver steatosis influenced the improvement of liver steatosis obtained by using a VLCKD. 

As far as we are aware, no information has ever been released as to whether the level of basic steatosis may influence the intensity of the effect of VLCKD in decreasing the quantity of liver fat in a population of subjects not taking any kind of drug. 

This study’s primary goal was to evaluate the impact of using a VLCKD for eight weeks on changes in liver steatosis and fibrosis, as measured by transient elastography (FibroScan), in 111 adults who were overweight or obese and were not suffering from any comorbid conditions. Modifications in anthropometric measurements (BMI, waist circumference [WC]), body composition (fat mass and fat-free mass, as determined by bioimpedance), and hormone and metabolic biomarkers (insulin, glucose, triglycerides, lipid composition, uric acid, and vitamin D) were also assessed. 

## 2. Materials and Methods

### 2.1. Study Design and Population

This 8-week real-life prospective study was conducted by our Center of Nutrition for the Research and the Care of Obesity and Metabolic Diseases of the National Institute of Gastroenterology at Saverio De Bellis Research Hospital (Castellana Grotte, Bari, Apulia, Italy). Ages between 18 and 64 were the primary requirements for inclusion, along with a BMI of at least 25 kg/m^2^. Assessments of anthropometric parameters through lab tests (biochemistry) were made, and the medical histories of all patients with overweight or obesity who made an appointment at our clinic were checked. The National and European guidelines’ list of contraindications for VLCKD was followed [27,28,30]. Based on the American and European recommendations, daily alcohol consumption was examined by asking patients directly about the number of glasses of alcohol they drank per day, as written in our previous studies. Patients who consumed more alcohol than recommended were not admitted. Every subsequent session began with a physical examination. The subjects were also questioned about smoking. The local Medical Ethics Committee approved the study protocol (Prot. n. 170/CE De Bellis). A total of 111 people took part in the study, which was conducted in compliance with the 1964 Helsinki Declaration. Prior to participating in the study, each subject gave their written consent. NCT05477212 is the study’s ClinicalTrials.gov identifier. Overweight or obese patients were enrolled from July 2022 to December 2023. Two check-ups were conducted: the first (T0) took place three weeks into the VLCKD treatment, and the second took place eight weeks following the VLCKD treatment (T1). Anthropometric measurements, fasting blood sample data, and results from instrumental testing (BIA and Fibroscan) were gathered at T0 and T1. A discussion of the study’s time scale can be found in earlier publications [34].

### 2.2. Diet Protocol

The VLCKD protocol used in the present study has been published in several previous studies of ours [33,34,36], and the first two steps were adopted on the basis of the description in the paper by Bruci et al. [29]. All participants received the VLCKD in accordance with the two-step protocol strategy from New Penta, Cuneo, Italy. According to the ketogenic nutritional therapy (KeNuT), a multi-step dietary model with meal replacements for the management of obesity and its related metabolic disorders [28], the VLCKD was a very low-calorie (650–800 kcal/day), low-fat (only 20 g per day), and low-carbohydrate (<30 g per day from vegetables) diet; the hallmarks of the initial phase of the VLCKD are based on olive oil. It has been established that 1–1.4 g of protein per kg of ideal body weight represents a fixed range per day that can maintain lean mass and meet the body’s minimum daily requirement. The participants were advised to drink at least two liters of water per day and consume less than eight hundred calories per day. Since the diet is out of balance and deficient in fresh fruit, micronutrient-enriched supplements were administered throughout the duration of the dietary treatment. During phase 1, patients ate high-biological-value protein replacement meals four or five times a day. Only vegetables with a low sugar content and low glycemic index were allowed to be eaten to ensure they received the recommended amount of fiber. During the active phase, extra virgin olive oil (two tablespoons per day), desired herbs and spices, and lemon were allowed. Phase 2 was comparable to the last phase, with a small increase in caloric consumption. To maintain nutritional ketosis, a natural protein meal was specifically reintroduced; in fact, the goal of this phase was to retrain the patient to eat fresh protein, gradually weaning them off of replacement meals. This phase was specifically divided into two sub-phases. In the first, we emphasized to the patients that dairy products were not yet permitted and substituted fresh protein from meat, fish, and eggs with a single-meal replacement (lunch or dinner). Fresh protein was substituted for both lunch and dinner during the second sub-phase [28].

### 2.3. Anthropometric Parameters

To determine body mass index (BMI) (kg/m^2^), measurements of height and weight were obtained while the subjects were barefoot, fasting, wearing light clothing, and had empty bladders. Every patient was measured using the same calibrated scale and stadiometer. The patients took off their outer garments and stood with their feet close together for us to measure their waist circumference (WC). The lower rib margin and the iliac crest were the two places where the circumference point was situated. The subject was seated while three separate measurements of the diastolic blood pressure (DBP) and systolic blood pressure (SBP) were made using an OMRON M6 automated blood pressure monitor. Every anthropometric measure was taken at baseline, three weeks into the VLCKD treatment, and eight weeks later.

### 2.4. Bioelectrical Impedance Analysis (BIA) 

Bioelectrical impedance analysis (BIA) was performed using a single-frequency bioimpedance analyzer (BIA-101 analyzer, 50-kHz frequency; Akern Bioresearch, Florence, Italy). A qualified nutritionist performed each measurement using established protocols. In compliance with the guidelines set forth by the European Society of Parenteral and Enteral Nutrition (ESPEN), the subjects underwent examination while lying supine with their legs separated [37]. They were instructed to abstain from food and drink for 12 h before the exam, as well as to refrain from exercising for 24 h. After the patients took off their shoes and socks, the contact areas were cleaned with alcohol right before the electrodes were attached. There have been prior descriptions of the injector and sensor electrode placement (BIATRODES Akern, Florence, Italy), as well as other BIA applications [37,38]. Akern software was used to calculate body composition parameters.

### 2.5. Biochemistry

Blood samples were drawn between 8:00 and 9:00 in the morning following an overnight fast. The concentrations of insulin, triglycerides, total cholesterol, LDL cholesterol, HDL cholesterol, AST, ALT, gamma-GT, and 25-OH vitamin D were measured, as well as fasting blood glucose, using the COBAS 8000 autoanalyzer (ROCHE Diagnostic SPA, Monza, Italy). The Capillarys 3 OCTA automatic capillary electrophoresis system (Sebia Italia S.r.l., Bagno a Ripoli, Firenze, Italy) was used to measure HbA1c. Insulin resistance was measured using the Homeostasis Model Assessment of Insulin Resistance (HOMA-IR) tool [39].

### 2.6. Liver Steatosis and Fibrosis Assessment by FibroScan

The level of liver fat was determined using the Fibroscan controlled attenuation parameter (CAP) (Echosens, Paris, France), which measures the degree of ultrasound attenuation caused by hepatic fat at the standard frequency of 3.5 MHz [40]. Studies carried out on obese patients suggest that liver biopsy and CAP have comparable diagnostic accuracy when it comes to determining the presence of hepatic steatosis. When liver stiffness values exceed 8 kPA, liver fibrosis is evident, and when CAP exceeds 275 dBm, mild MASLD is evident [41].

### 2.7. Statistical Analysis

The described variables, pre-(T0) and post-(T1) VLCKD, are reported as means (SD) or medians (IQR).

The baseline variables, which are continuous variables expressed as mean ± Standard Deviation (SD) and median (IQR), were statistically analyzed. Shapiro’s test was used to determine whether each variable distribution was normal. 

Using Wilcoxon’s rank sum test for paired samples, any statistical differences between the pre- and post-nutritional treatment states were evaluated. 

Also, 95% Confidence Intervals (CIs) for *p*-values less than or equal to 0.05 were used to determine statistical significance.

The differences between T1 and T0 for CAP and liver stiffness were found to be normally distributed, so linear regression was performed in the models. For each variable, the normal distribution was ascertained using Shapiro’s test [42].

CAP T0 was used as the regressor, alongside CAP difference as the dependent variable, in the original linear regression model (*model a*). In the second linear regression, *model a* was adjusted for CAP T0, sex, age, WAIST T0, Systolic BP T0, Diastolic BP T0, Homa Index T0, HbA1c T0, Triglycerides T0, Total Cholesterol T0, and ALT T0 as dependent variables (*model b*). The same analysis was carried out for liver stiffness. 

Statistical analysis was conducted using Stata statistical software version 18.0 (StataCorp, 4905 Lakeway Drive, College Station, TX, USA) [43].

## 3. Results

### 3.1. The Study Population’s Baseline Characteristics and Changes Following the VLCKD

Overall, 67.57% of those surveyed in the study were women (N = 75/111), 85.59% were obese (N = 95/11), and 22.52% were smokers (N = 25/111). The participants’ ages ranged from 18 to 64 years old, while the mean age was 41.89 years (±12.66) (Table 1).

### 3.2. Changes in Clinical and Laboratory Parameters after the VLCKD

Table 2 shows the significant changes in every parameter after eight weeks of the VLCKD treatment. Regarding liver parameters, following the VLCKD, there was a significant decrease in the CAP (the fibroscan parameter of steatosis). It appears that the diet decreased the levels of γGT and ALT in the blood. At variance with CAP, hepatic stiffness (the fibroscan parameter of fibrosis) showed only a tendency to a decrease after the VLCKD.

The variables that showed the reductions observed after the VLCKD included reductions in body mass index, fat mass, waist circumference, insulin resistance (assessed by HOMA-IR), HbA1c, fasting blood glucose, insulin, total cholesterol, LDL cholesterol and HDL cholesterol, triglycerides, and systolic and diastolic blood pressure. On the other hand, following the VLCKD, the circulating levels of vitamin D were higher (Table 2).

The relationships between the ∆ (delta = ∆) of CAP and CAP T0 in a raw model and multivariate models are presented in Table 3.

The CAP delta value was positively and statistically significantly correlated with CAP T0 (before VLCKD) in both models: model a (0.37 95% CI 0.21; 0.52) and the multivariate model (*model b*: 0.41 95% CI 0.20; 0.62) (Table 3 and Figure 1).

Table 4 shows the associations between the ∆ (delta = ∆) of liver stiffness and liver stiffness T0 in a raw model and multivariate model. 

∆ liver stiffness was also found to be positively and statistically significantly correlated with liver stiffness before the VLCKD treatment (T0) in both models, with the value for model a being 0.52 (95% CI 0.39; 0.65) and that for the multivariate model (model b) being 0.46 (95% CI 0.29; 0.63) (Table 4 and Figure 2).

## 4. Discussion

The purpose of this study was to assess whether a higher level of steatosis may positively or unfavorably influence the effectiveness of a VLCKD in improving steatosis and fibrosis among patients who suffer from obesity and overweight. A total of 75 women and 36 men, ages 18 to 64, who were either overweight (14.41%) or obese (85.59%) and did not use drugs were the subjects of an investigation.

This study supports our earlier research [32,33] by showing a significant decrease in the CAP, a FibroScan parameter that measures the accumulation of fatty liver, after the VLCKD treatment. In particular, the research demonstrates that subjects with higher values of CAP, corresponding to a condition of increased liver steatosis, obtain a greater reduction in liver steatosis after 8 weeks of VLCKD treatment (800 kcal/day). 

Due to the hepato-protective effects of carbohydrate restriction, which are further enhanced by a lower total calorie intake and ketogenesis, we proposed in a previous study that reducing daily carbohydrate intake to less than 50 g, which induces ketosis, could improve MASLD [31]. Furthermore, it remains plausible that the reduction in steatosis resulting from the use of the VLCKD may be partially attributed to the anti-inflammatory properties of the VLCKD itself [34]. According to this theory, we recently showed that patients with steatosis have lower WBC and platelet counts, while those without steatosis (CAP ≤ 275 dB/m) did not exhibit a significant decrease in WBC and platelet counts [36]. This suggests that subjects with liver steatosis may be more susceptible to the potential anti-inflammatory effect of VLCKD than those without this condition. Lastly, it may well be that the marked decrease in insulin levels and/or in insulin resistance has a role in accelerating the decrease in liver steatosis. All these hypotheses need to be confirmed by further intervention studies. 

The only finding regarding liver stiffness and the VLCKD was a tendency for a reduction in the FibroScan parameter, which measures fibrosis in the liver. It may well be that the lack of a significant influence of the VLCKD on fibrosis in this study is because few patients had liver fibrosis (stiffness values > 8 kPA using fibroscan) before the dietetic treatment. However, interestingly, subjects with higher basal values of kPA obtained a greater reduction in liver fibrosis after 8 weeks of VLCKD treatment, confirming that subjects who benefit more from VLCKD treatment are those with worse steatosis and fibrosis at the moment of enrollment. 

Most of the findings from this study regarding the effects of VLCKD in obese subjects are confirmed by our previous study [33]. In terms of anthropometric measurements, VLCKD treatment significantly reduced BMI, WC, fat mass, and blood pressure at both the systolic and diastolic levels. In terms of metabolic parameters, VLCKD treatment effectively reduced fasting blood glucose, insulin, triglycerides, total cholesterol, LDL cholesterol, HDL cholesterol, γGT, and insulin resistance. The notion that VLCKD treatment considerably increases insulin sensitivity is supported by the majority of these results [33]. At variance with the decreases in most of the investigated variables, VLCKD treatment induced an increase in vitamin D levels. The hypothesis that vitamin D is stored in adipose tissue and released after weight loss is supported by the increase in vitamin D after significant weight loss [43].

### Strength and Limitations

This study’s primary advantage is its sample size: 111 subjects were examined to determine the effects of the VLCKD. Furthermore, in order to rule out any possible interactions, the study was limited to participants from southern Italy who had similar profiles and who were not taking any dietary supplements or pharmaceutical therapies. This provides methodological validity to the study. The prospective nature of this study allowed us to determine the temporal nature of the connections and their linkages. FibroScan is the only method that guidelines recommend using to assess hepatic steatosis when a biopsy is not feasible, and it is commonly used in cases of suspected liver steatosis. The absence of a control group in the study design represents a significant drawback and limitation as it prevented any meaningful comparison with a patient group following a very low-calorie balanced diet or one with a high carbohydrate content. 

## 5. Conclusions

We found that a significant decrease in liver steatosis occurs after treating individuals with overweight and obesity with the VLCKD, as shown by a significant decrease in CAP, investigated by FibroScan, and that this improvement is significantly higher in subjects with a greater level of steatosis before the diet. At variance with the steatosis result, VLCKD was followed by a non-significant trend towards a decrease in liver fibrosis. 

## Figures and Tables

**Figure 1 nutrients-16-00874-f001:**
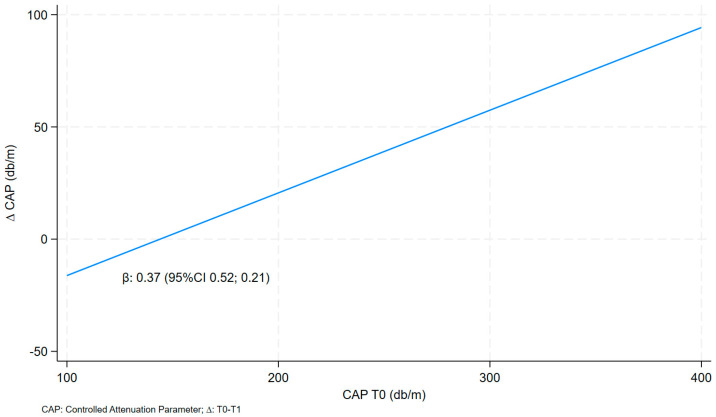
Linear fit prediction for delta CAP on CAP T0.

**Figure 2 nutrients-16-00874-f002:**
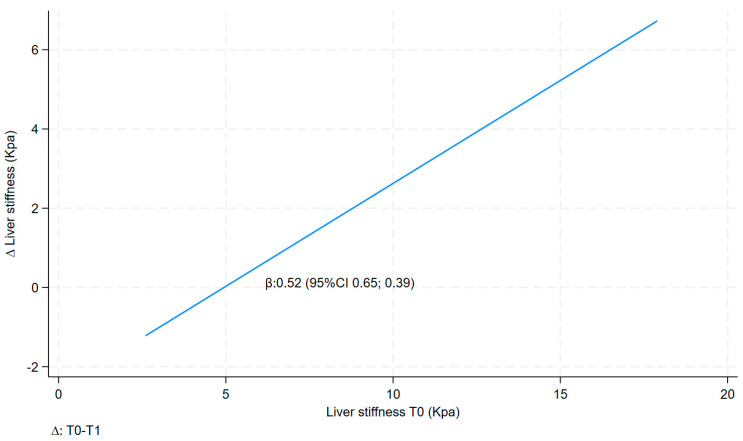
Linear fit prediction for delta liver stiffness on liver stiffness T0.

**Table 1 nutrients-16-00874-t001:** Baseline characteristics.

	Pre VLCKD
N	111
Age	41.89 (±12.66)
Sex	
Female	75 (67.6%)
Male	36 (32.4%)
Smoke	
Never/former	86 (77.5%)
Current	25 (22.5%)

**Table 2 nutrients-16-00874-t002:** Description of the entire sample during the evaluation periods (pre- and post-diet). All data are displayed as median (IQR) or mean (+SD).

	Pre-VLCKD	Post-VLCKD	*p*-Value *
	Mean (SD)	Mean (SD)	
N	111	111	
Body Mass Index (kg/m^2^)	35.66 (6.08)	32.56 (5.72)	<0.001
Waist Circumference (cm)	111.24 (13.71)	102.42 (13.42)	<0.001
Systolic BP (mmHg)	130.11 (12.27)	123.18 (9.09)	<0.001
Diastolic BP (mmHg)	82.18 (9.91)	76.47 (7.17)	<0.001
FBG (mg/dL)	93.75 (9.57)	88.26 (9.78)	<0.001
Insulin (μU/mL)	16.71 (10.09)	9.80 (4.87)	<0.001
HOMA index	3.88 (2.41)	2.17 (1.14)	<0.001
HbA1c (%)	5.44 (0.38)	5.24 (0.35)	<0.001
Triglycerides (mg/dL)	108.96 (62.22)	84.16 (35.56)	<0.001
Cholesterol (mg/dL)	194.53 (46.26)	167.44 (38.53)	<0.001
HDL Cholesterol (mg/dL)	53.16 (14.44)	46.88 (11.23)	<0.001
LDL Cholesterol (mg/dL)	131.74 (33.70)	110.76 (29.02)	<0.001
25-OH-Vitamin D (ng/mL)	21.30 (6.87)	26.52 (7.64)	<0.001
AST (U/L)	21.49 (10.64)	19.36 (6.62)	0.28
ALT (U/L)	29.62 (23.65)	23.48 (14.56)	0.031
γ-GT (U/L)	24.84 (16.68)	16.17 (8.50)	<0.001
CAP (db/m) ^#^	279 (244; 325)	232 (188; 277)	<0.001
Liver Stiffness (Kpa) ^#^	5.20 (4.20; 6.30)	5.10 (3.80; 6.10)	0.33
Fat Mass (Kg)	39.48 (12.58)	32.83 (11.42)	<0.001
Fat-Free Mass (kg)	58.19 (11.69)	56.03 (11.12)	0.095

* Wilcoxon’s rank sum test for paired samples; ^#^ median (IQR); VLCKD: very low-calorie ketogenic diets, BP: blood pressure, FBG: fasting blood glucose, 25-OH-vitamin D: 25-hydroxyvitamin D, AST: Aspartate Amino Transferase, ALT: Alanine Transaminase, γGT: gamma-glutamyl transpeptidase, CAP: controlled attenuation parameter.

**Table 3 nutrients-16-00874-t003:** Linear regression model on delta CAP (T0 − T1) as dependent variable.

∆ CAP	Β	95% CI
*Mode a*		
CAP T0	0.37 *	0.21; 0.52
*Model b*		
CAP T0	0.41 *	0.20; 0.62

*Model a*: adjusted for CAP T0. *Model b* adjusted for CAP T0, sex, age, WAIST T0, Systolic BP T0, Diastolic BP T0, Homa Index T0, HbA1c T0, Triglycerides T0, Total Cholesterol T0, and ALT T0. * *p*-value < 0.001. CAP: controlled attenuation parameter; ∆: T0 − T1.

**Table 4 nutrients-16-00874-t004:** Linear regression model on delta liver stiffness (T0 − T1) as dependent variable.

∆ Liver Stiffness	Β	95% CI
*Model a*		
Liver stiffness T0	0.52 *	0.39; 0.65
*model b*		
Liver stiffness T0	0.46 *	0.29; 0.63

*Model a*: adjusted for liver stiffness T0, *Model b* adjusted for liver stiffness T0, sex, age, WAIST T0, Systolic BP T0, Diastolic BP T0, Homa Index T0, HbA1c T0, Triglycerides T0, Total Cholesterol T0, and ALT T0. * *p*-value < 0.001; ∆: T0 − T1.

## Data Availability

The article contain the original contributions that were presented in the study. The corresponding author can be contacted for any additional questions.

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
