# Peer review of "Higher-Level Steatosis Is Associated with a Greater Decrease in Metabolic Dysfunction-Associated Steatoic Liver Disease after Eight Weeks of a Very Low-Calorie Ketogenic Diet (VLCKD) in Subjects Affected by Overweight and Obesity"

_nutrients, 2024, doi:10.3390/nu16060874_

Round 1

Reviewer 1 Report

Comments and Suggestions for Authors

Thank you for submitting the manuscript "Higher Level Steatosis is Associated to a Greater Decrease of Steatosis After Eight Weeks' Very Low-Calorie Ketogenic Diet (VLCKD) in Subjects Affected by Overweight and Obesity and Metabolic Dysfunction-Associated Steatotic Liver Disease (MASLD)" to Nutrients. Overall, the biggest problem with this manuscript is that it does not have a control group, that is, only the VLCKD intervention was evaluated, but it is not known whether it is better or worse than a balanced diet where the percentage of carbohydrates is higher. This needs to be very well clarified in the limitations of the work.

- The title is too long and makes it difficult to understand. Needs to be improved.

- The introduction is very well written and offers enough information to understand the manuscript.

- Lines#148-150: describe how much each macronutrient contributes to the total energy value of the diet.

- Furthermore, I found the data still a little incipient to answer the work hypothesis as it only involved 8 weeks of study. Does this manuscript report only part of the data and is the study still ongoing?

Comments on the Quality of English Language

 Minor editing of English language required.

Reviewer 2 Report

Comments and Suggestions for Authors

This interesting prospective study by Sila et al. showed that a significant decrease in liver steatosis by CAP occurred after treating individuals with overweight and obesity with VLCKD, and that this improvement was significantly higher in subjects with a greater level of steatosis before the diet. However, there was no significant trend to a decrease of liver fibrosis after VLCKD diet. However, there are still some major concerns to be addressed:

1. A new terminology of MASLD has been raised to represent NAFLD in AASLD 2023 conference (https://www.aasld.org/new-masld-nomenclature). MASLD encompasses patients who have hepatic steatosis and have at least one of five cardiometabolic risk factors but without significant alcohol use. All the participants in this study with fatty liver and overweight/obesity, which is accordance with the definition of MASLD. Therefore, please change all the terminology of NAFLD to MASLD.

2. It is too simple to describe baseline characteristics in this paper. Therefore, please add one Table for baseline characteristics including all the characteristics also in Table 2.

3. There is no detail introduction of VLCKD diet protocol but just citation for previous paper. Could you summarize and describe the VLCKD diet protocol in more detail in this paper?

4. For Table 3 and 4, in linear regression on Delta CAP and LSM, the adjusted variables were not enough in the models. For model a, please adjust age, gender, BMI and Waist; For model b, please adjust age, gender, BMI, Waist, blood pressure, fasting glucose, HbA1c, TG, TC and ALT, which includes factors associated with comorbidities of NAFLD.

5. You mentioned the participants were questioned about alcohol use and smoking in this study. Could you also show the data of alcohol use and smoking in the Baseline characteristics Table and Table 2 for comparison.

Round 2

Reviewer 2 Report

Comments and Suggestions for Authors

The authors fully addressed my comments.